# Best Supportive Care of the Patient with Oesophageal Cancer

**DOI:** 10.3390/cancers14246268

**Published:** 2022-12-19

**Authors:** Rita Carrilho Pichel, Alexandra Araújo, Vital Da Silva Domingues, Jorge Nunes Santos, Elga Freire, Ana Sofia Mendes, Raquel Romão, António Araújo

**Affiliations:** 1Department of Medical Oncology, Centro Hospitalar Universitário do Porto (CHUPorto), 4099-001 Porto, Portugal; 2Department of Internal Medicine, Centro Hospitalar Universitário do Porto (CHUPorto), 4099-001 Porto, Portugal; 3Instituto de Ciências Biomédicas Abel Salazar (ICBAS), Universidade do Porto (UP), 4050-346 Porto, Portugal; 4Department of General Surgery, Centro Hospitalar Universitário do Porto (CHUPorto), 4099-001 Porto, Portugal; 5Equipa Intra-hospitalar de Suporte em Cuidados Paliativos, Centro Hospitalar Universitário do Porto (CHUPorto), 4099-001 Porto, Portugal; 6Unidade Multidisciplinar de Investigação Biomédica (UMIB), 4050-346 Porto, Portugal

**Keywords:** palliative supportive care, oesophageal cancer, dysphagia, malnutrition, oesophageal fistula

## Abstract

**Simple Summary:**

Oesophageal cancer is the sixth leading culprit of cancer-related mortality, and the majority of the patients with advanced disease are treated with best supportive care intent. Several management alternatives have been developed in recent years to address palliation of oesophageal cancer, such as self-expandable metallic stent placement and hypofractionated radiotherapy. Yet, optimal management is not standardized, and best supportive care decisions should be discussed on a case-by-case basis by multidisciplinary teams. This evidence-based review aimed for defining recommendations on the management of oesophageal cancer main symptoms and complications, such as dysphagia, malnutrition, pain, nausea and vomiting, fistula and bleeding. The late goal of our review is to improve (toward the “best”) supportive care and decision making for oesophageal cancer patients.

**Abstract:**

Background: Oesophageal cancer patients have poor survival, and most are unfit for curative or systemic palliative treatment. This article aims to review the best supportive care for oesophageal cancer, focusing on the management of its most frequent or distinctive symptoms and complications. Methods: Evidence-based review on palliative supportive care of oesophageal cancer, based on Pubmed search for relevant clinical practice guidelines, reviews and original articles, with additional records collected from related articles suggestions, references and societies recommendations. Results: We identified 1075 records, from which we screened 138 records that were related to oesophageal cancer supportive care, complemented with 48 additional records, finally including 60 records. This review summarizes the management of oesophageal cancer-related main problems, including dysphagia, malnutrition, pain, nausea and vomiting, fistula and bleeding. In recent years, several treatments have been developed, while optimal management is not yet standardized. Conclusion: This review contributes toward improving supportive care and decision making for oesophageal cancer patients, presenting updated summary recommendations for each of their main symptoms. A robust body of evidence is still lacking, and the best supportive care decisions should be individualized and shared.

## 1. Introduction

Worldwide, oesophageal cancer is the eighth-most common cancer and the sixth leading cause of cancer-related mortality [1,2]. In Portugal and in Europe, it is far less incident, being the twentieth-most common cancer and the thirteenth leading cause of cancer-related mortality [1,3]. It remains, however, a highly fatal disease, with a 5-year overall survival rate of 20% for all stages, and 5% if considered metastatic disease alone [4].

From 1990 to 2017, there was an important decrease in age-standardized incidence (22.0%), mortality (29.0%), and disability-adjusted life years (33.4%) globally [2]. Still, the prognosis of oesophageal cancer remains poor. Oesophageal cancer is usually diagnosed late in the course of the disease and patients often have other relevant comorbidities, rendering them unfit for curative modalities of treatment [5].

Current guidelines on oesophageal cancer still suggest that, if there are distant metastases, or if the tumour is unresectable/invades adjacent structures/is too extensive and cannot be treated with surgery or curative-intent chemoradiotherapy, the patient should be treated with palliative intent, aimed at relieving cancer-related symptoms, improving quality of life (QOL) and prolonging survival. A poor performance status or relevant comorbidities preclude systemic treatment, so that only one fifth of incurable gastric or oesophageal cancer patients receive palliative chemotherapy (ChT) [6,7,8]. The advent of immunotherapy and targeted therapies applied to digestive tract tumours may allow us to treat more patients with advanced oesophageal cancer. Yet, for the majority of them, the most appropriate option remains the best supportive care.

Therefore, and trying to address the lack of therapeutic guidance [9], and our own clinical practice challenges, we aim to review the best palliative supportive care of oesophageal cancer, focusing on the management of its most frequent or distinctive symptoms and complications.

## 2. Methods

Authors performed an evidence-based review on palliative supportive care of oesophageal cancer, searching for relevant clinical practice guidelines, systematic reviews and the latest original articles. Our main search strategy included searching MEDLINE via PubMed, with the following search headings: ((supportive care) OR (palliative)) AND ((oesophag *) OR (esophag *)) AND ((cancer) OR (carcinoma)). Pubmed search was last undertaken on 25 August 2022. We also performed a secondary search, via PubMed, Cochrane and societies’ sites, searching for “per symptom” epidemiology and treatment. We have also included articles collected from related articles suggestions and references sections. Selection flowchart is represented on Figure 1.

## 3. Results and Discussion

Main database search returned 1075 records, complemented with 63 additional records from “per symptom” searching, related articles, references and societies resources. From these, we selected 153 records that were related to oesophageal cancer supportive care, from which we included 70 in the qualitative synthesis.

### 3.1. Management of Oesophageal Cancer Symptoms or Complications

Mostly from empirical knowledge, we may say that dysphagia, weight loss and malnutrition, chest pain, nausea and vomiting, digestive bleeding and fistulisation to cardiopulmonary system are some of the main symptoms or complications related to oesophageal cancer, which we addressed in the following sections. Table 1 presents a summary of recommendations as explained on the text.

#### 3.1.1. Dysphagia or Obstruction

Malignant dysphagia is the predominant symptom in more than 70% of patients with oesophageal cancer [10]. Usually, it arises from oesophageal lumen obstruction, but it can also be caused by tumour related dysmotility.

Regarding dysphagia palliation, there is no standard of care. Choosing the best treatment modality for each patient must rely on a multidisciplinary discussion, considering the symptoms burden and vital prognosis. In general, the first option for immediate relief of dysphagia is the insertion of a (fully or partially)-covered self-expandable metallic stent (SEMS). Palliative radiotherapy (RT) is another common option, more appropriate for patients with a longer life-expectancy, greater than three months [11,12,13].

SEMS placement is more effective and quicker in palliating dysphagia compared to other endoscopic procedures, which is of particular interest in cases of severe dysphagia and short life expectancy [14]. Complications from oesophageal stents are not negligible [15]. Different stents have been developed and were compared elsewhere [16].

Palliative RT, when compared to stent placement, has been shown to improve survival and QOL, and is being increasingly adopted [14,17]. Additionally, it correlates with significantly lower risk of toxicity, better pain control and equivalent relief of moderate to severe dysphagia [18]. Although intraluminal brachytherapy proved to be effective [14], it is more invasive and less widely implemented than external beam RT [9,19,20], with comparable efficacy [21]. Furthermore, patient-reported outcomes seem to favour external beam RT [22]. Short course palliative RT (20Gy in 5 fractions in consecutive workdays) was better tolerated with equal palliative effects than longer course [23].

Although it was theoretically promising, a recent trial failed to show additional benefit from adding palliative RT to SEMS insertion [24].

Laser ablation and photodynamic therapy require greater expertise than SEMS placement and more often need reintervention. Rigid plastic tube insertion and dilatation, either alone or in combination with other modalities, are no longer recommended due to high rate of delayed complications and recurrent dysphagia [14].

Complementarily, it is important to control pain, such as odynophagia, and also to educate patients for dietary changes, such as smaller and more frequent meals, softer food, liquid diets and oral nutritional supplements.

#### 3.1.2. Malnutrition

Malnutrition (as synonym of undernutrition) and caquexia (as synonym of chronic disease-related malnutrition with inflammation) are common in advanced cancer disease and lead to poor QOL and performance status and to decreased survival [25,26,27,28]. Cancer caquexia is defined by either weight loss >5% alone, or weight loss >2% if body mass index is <20 kg/m^2^ or fat free mass is reduced [29].

Patients with cancer of the upper digestive tract are prone to malnutrition, often present at diagnosis [26]. Oesophageal cancer raises specific nutritional concerns, as the disease predisposes to dysphagia, cancer-related caquexia and invasive therapies.

More recently, studies and best-practice guidelines increasingly advocate for early identification of malnutrition and implementation of nutrition interventions, aimed at maintaining or improving QOL [30,31,32,33]. These may also increase survival [34]. However, when life expectancy is shortened to few months or weeks, the intensity of such nutritional interventions should be decreased and focus should be directed towards immediate symptomatic relief and patient comfort (addressing thirst, eating-related distress, favourite foodstuffs and flavours) while dealing with family or carer concerns on end-of-life nutrition and hydration [33,35].

Since it is largely underdiagnosed and undertreated, routine nutritional status should be assessed with appropriate screening tools. The Patient-Generated Subjective Global Assessment (PGSGA), the Subjective Global Assessment (SGA), and Nutrition Risk Index (NRI) are examples of malnutrition screening tools validated for cancer patients with adequate specificity and sensitivity [31].

Oesophageal obstruction treatment as described above should be attempted, since even little oral feeding can achieve better QOL compared to enteral feeding. Nutritional counselling and support remains a cornerstone of successful management of patients with incurable oesophageal cancer, maintaining weight and performance status, even in the elderly. Clinical assisted nutrition, such as enteral feeding and parenteral nutrition should be based on a careful evaluation of the potential benefits, as QOL and functional expectant gain, and on the patient’s and family’s wishes [32,36,37]. Specifically about parenteral nutrition, it may benefit a limited percentage of patients when oral or enteric feeding is not possible, allowing them to survive longer [38,39]. When available, home parenteral nutrition allows good acceptance and stable QOL [39]. The decision to discontinue should attend the predetermined goals for each case and must be reconsidered in end-of-life situations.

#### 3.1.3. Pain

Pain is one of the most common cancer-related symptoms and has a significant negative impact on QOL. Almost every patient with oesophageal cancer will experience pain at some point of the disease course, particularly in end stage disease [40].

The pain of oesophageal cancer can arise from different aetiologies, such as direct compression or invasion from tumour, and effects from treatments. Pain from the oesophagus usually manifests as odynophagia (with swallowing and eating), or chest and back pain at rest [40].

Following a stepwise approach, initial management with paracetamol (acetaminophen) is recommended for patients presenting with mild pain, although its effectiveness for cancer pain was not proven [41,42]. Non-steroidal anti-inflammatory drugs (NSAID) can also be considered, but their clinical benefit should be assessed within two weeks and balanced with the risks of gastroduodenal and cardiovascular toxicity [43,44]. Selective COX-2 inhibitors may be preferred to NSAID in this population. Metamizole (dipyrone) is an alternative non-opioid analgesic drug [45]. For moderate or severe cancer pain, opioids are the mainstay of pain management [46,47]. It is recommended to start with as-needed dosing of a short-acting opioid. Opioid dosing and titration should follow cancer-pain guidelines issued by World Health Organization and by associations dedicated to the management of this matter [42,44]. In the setting of dysphagia or odynophagia, liquid formulations or sublingual preparations of short-acting opioids should be preferred. Regarding long-acting (sustained or extended-release) opioids, there are few alternatives to tablets, such as fentanyl and buprenorphine transdermal systems, and some specific oxycodone and morphine sulfate capsules that can be opened so the content can be swallowed or administered through a feeding tube [40]. Yet, these options are not available in every country or institution. Alternatively, subcutaneous morphine may be administered by a portable pump, in perfusion or boluses (in patient-controlled analgesia systems). When prescribing opioids, patients have to be educated on the expected side effects, such as opioid-induced constipation, and prophylactic laxatives, such as senna and polyethylene glycol, should be prescribed [40,44,48].

Oesophageal cancer patients may also experience neuropathic pain, from tumour invasion of nerve plexus or as a side effect of platinum or taxane-based ChT previously used. For these patients, use of gabapentin, pregabalin, duloxetine or venlafaxine may be helpful [40,44]. When inflammatory pain is present, corticosteroids may also be used for pain control.

RT is also an effective palliative treatment for localized pain from the primary tumour or its metastasis. Hypofractionated RT (specifically 8Gy single dose) results in excellent pain relief from bone metastasis, and re-irradiation can be effective in recurrent pain [44,49]. The best response to pain relief occurs two to four weeks after completion of radiation, so, this treatment should be coupled with medical management and reserved to patients who have an expected survival of several weeks to months, in order to benefit from it [40].

Recent evidence favours the use of acupuncture and acupressure, as they significantly reduced pain intensity and opioid use [50].

#### 3.1.4. Nausea and Vomiting

Nausea in patients with oesophageal cancer can have many causes, including mechanical obstruction, mucositis from ChT and radiation, pain medication, dehydration and anxiety [40].

For most advanced cancer patients, the best antiemetic choice are prokinetic drugs such as metoclopramide [51,52]. Alternative sedative antiemetic drugs include haloperidol, chlorpromazine or olanzapine. Anxiolytics are helpful for anxiety-driven nausea. Corticosteroids are synergistic with metoclopramide and 5-HT3 antagonist against chronic nausea, and may be most useful in nausea and vomiting induced by intracranial disease or bowel obstruction. Whenever feasible, mechanical obstruction nausea is more effectively managed with interventional procedures [40,52].

#### 3.1.5. Fistula

The unique anatomy of the oesophagus and close proximity to the cardiopulmonary system favours the risk of major morbidity from cancer. Oesophageal fistula develops in 5 to 20% of oesophageal cancers, most commonly between the oesophagus and the respiratory tract (Figure 2a), and occasionally to pleural space, the aorta or other mediastinal structures (Figure 2b). Apart from tumour invasion, fistulas may also develop secondarily to radiation or endoscopic therapies. A history of worsening dysphagia and dyspnea, and coughing temporally related to drinking and eating is highly suggestive of an oesophagorespiratory fistula [53,54].

Oesophagorespiratory fistula are life-threatening, and immediate treatment should be attempted, as they can lead to aspiration pneumonia and poor nutritional intake, resulting in poorer survival (1 to 6 weeks with supportive care alone) [55].

SEMS placement is the treatment of choice for an oesophagorespiratory fistula [56]. In most published reports, complete sealing of the fistula was established in 90% of patients, while only half of these patients achieve long-term fistula closure [53,57,58]. In the palliative setting, surgical options, such as closure or resection of the fistula or bypass surgery, are associated with high morbidity and mortality and are not recommended [59].

#### 3.1.6. Bleeding and Anaemia

Gastrointestinal (GI) bleeding in the cancer patient is a common and challenging clinical problem. Optimal clinical care is granted by access to a multidisciplinary team, including a gastroenterologist, an interventional radiologist, a radiation oncologist and a surgeon. It widens treatment options, to reach the most appropriate solution for each case [60].

The oesophagus is in close contact with the aorta and its arterial blood supply arises mainly from aortic small branches, but also from left gastric artery branches to the distal oesophagus [61].

In advanced oesophageal cancer, arterial bleeding can be massive and life-threatening, specially arising from aortic fistula. For instance, advanced care planning must include scope of treatment orders in such scenario: either to adopt an emergent, aggressive and invasive potentially life-saving treatment, or to allow natural death and improve comfort, including palliative sedation.

Endoscopic therapies are well established treatments of GI bleeding, but have been shown to be less effective to control malignancy-related bleeding [60,62]. Nevertheless, endoscopy is preponderant for diagnosing the source of bleeding and endoscopic interventions may avert the need for emergent surgery or temporarily control bleeding as a bridge for treatment with RT. Reflecting the lack of data on this matter, there are no widely accepted guidelines for optimal endoscopic therapeutic approach for tumour bleeding [60]. One of the caveats is that often tumour haemorrhage is diffuse and local endoscopic therapy fails to control the whole of it. Argon plasma coagulation is the method that has more evidence supporting its use, yet the majority of data indicate that control of bleeding is not reliably achieved [63]. Hemospray is a promising haemostatic agent for endoscopic application: for upper GI bleeding related to malignancy, the overall initial haemostasis rate was 94.9%, and the rebleeding rate was 30.3% [64].

RT has a pivotal role in the treatment of GI tumours. However, data describing RT for the treatment of bleeding are limited for oesophageal cancer [65]. In the setting of chronic blood loss, RT is suitable and may provide benefit as shown for gastric carcinoma. Common fractionation schedules use 1, 5 or 10 fractions, according to the expert radiation oncology physician judgment and discretion, based on many factors [66]. In case of active hemodynamically significant haemorrhage, RT is not useful for initial emergent bleeding control but it may latter complement the endoscopic or angiographic interventions.

Angiography and arterial embolization are feasible for the territory of the celiac artery and its branches. Here, the left gastric artery, which provides branches to distal oesophagus, is of particular interest. Considering oesophageal anatomic relations and blood supply, angiographic therapy can be only attempted in distal oesophagus bleeding as rescue, when endoscopic treatment fails. Yet, data supporting its use are limited.

Anaemia can adversely affect health related QOL of many cancer patients. Oesophageal cancer anaemia may particularly arise from tumour bleeding or malnutrition (addressed above). After assessing and treating iron deficiency or other reversible causes of anaemia, red blood cell transfusion can be used to treat symptomatic anaemia, as part of best supportive care [67]. Erythropoiesis stimulating agents are not recommended in patients who are not on ChT [68]. General recommendations on cancer anaemia management and palliative care experts’ advisory can be applied to oesophageal cancer patients, taking into account their personal characteristics and options.

### 3.2. Special Considerations

Evidence relating specifically to elderly patients is limited, but available data suggest they derive similar benefit from treatment as younger patients [69].

Continuous updating of clinical management of patients with oesophageal cancer demands specialized multidisciplinary teams, including surgical, medical and radiation-oncology specialists, well as gastroenterologists, palliative care specialists, nutritionists, and nurses [7,70]. Referral to a palliative care specialist is recommended for oesophageal cancer patients with symptoms refractory to initial management and any cases that are particularly complex or difficult to manage.

## 4. Conclusions

Oesophageal cancer has a great personal and global impact. Often it presents as an advanced disease, with patients requiring best supportive care. However, apart from trials on the use of stents, there is a lack of high-level evidence regarding the several treatment options available. More well-designed trials and observational studies are needed to determine which approaches achieve the greatest benefit for these patients. Optimal management is not yet standardized, and best supportive care decisions should be discussed on a case-by-case basis by multidisciplinary teams.

## Figures and Tables

**Figure 1 cancers-14-06268-f001:**
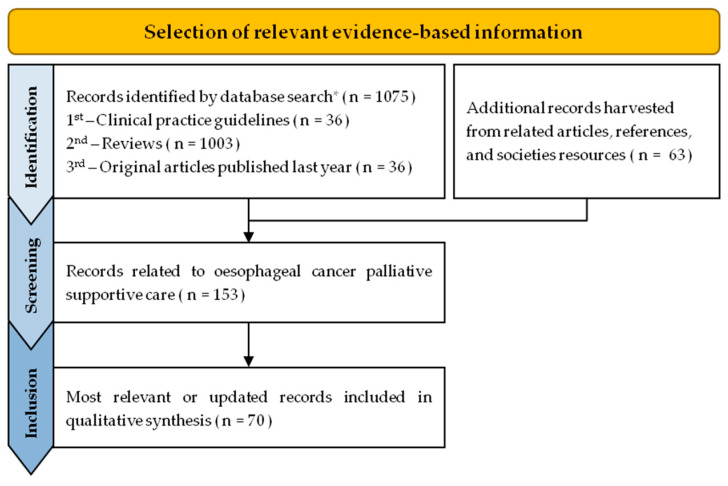
Flowchart of selection of relevant evidence-based information on palliative supportive care of oesophageal cancer. * PubMed search heading: ((supportive care) OR (palliative)) AND ((oesophag *) OR (esophag *)) AND ((cancer) OR (carcinoma)), last executed on 25 August 2022.

**Figure 2 cancers-14-06268-f002:**
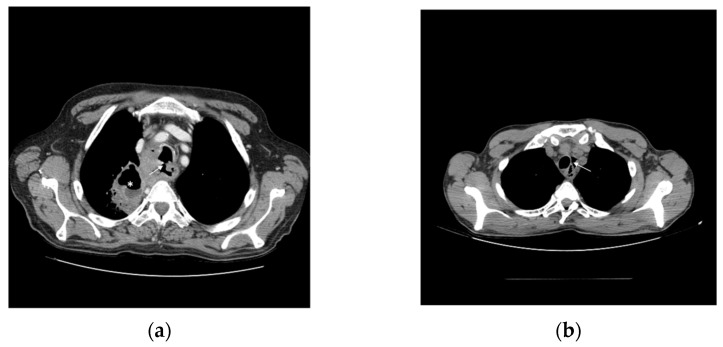
Oesophageal fistula. (**a**) Tracheo-oesophageal fistula (arrow) associated with pneumonia and lung abscess (*) (**b**) Squamous carcinoma of cervical oesophagus complicated with pneumomediastinum (arrow).

**Table 1 cancers-14-06268-t001:** Summary of recommendations.

	Dysphagia	Malnutrition	Pain	Nausea and Vomiting	Fistula	Bleeding and Anaemia
*Do*	Hypofractioned EBRT Brachytherapy Covered SEMS placement Pain control Dietary changes Adequate drug formulations (ex: liquid, powder, td, transmucosal, sc)	Malnutrition screening Early nutrition intervention Dysphagia treatment As-needed enteral tube feeding and parenteral nutrition (if expected survival of months) Assessment of end-of-life concerns on nutrition and hydration	(Re-)Assessment of the pain cause and its characteristics Stepwise approach Paracetamol ^a^ NSAID ^a,b^ Opioids (preferred oral morphine, or td fentanyl; or iv route for rapid pain control) Neuropathic-pain agents Corticosteroids Palliative antalgic RT Acupuncture and acupressure	Metoclopramide Sedative antiemetic drugs (ex: haloperidol, olanzapine) 5-HT3 agonists Anxiolytics Treat mechanical obstruction, if feasible Dexametasone, as part of the treatment of brain metastases or malignant bowel obstruction	Immediate treatment SEMS placement	Hemospray Palliative haemostatic RT Advanced care planning for the scenario of massive bleeding Assessment and treatment of causes of anaemia Red blood cell transfusion for symptomatic anaemia
*Don’t*	Rigid plastic tube insertion Dilatation Laser ablation Photodynamic therapy	End-of-life invasive clinical assisted nutrition	Forget about side effects		Surgery in the palliative setting	RT for massive bleeding (due to great vessels fistula) Erythropoiesis stimulating agents
*Don’t know*	RT and SEMS combination			Add dexametasone ^c^		Argon plasma coagulation Arterial embolization

EBRT: external bean radiotherapy; NSAID: Non-steroidal anti-inflammatory drugs; RT: radiotherapy; SEMS: self-expandable metal stent; sc: subcutaneous; td: transdermal. ^a^ First step of the WHO analgesic ladder and widely used, despite no proven effectiveness for cancer pain. ^b^ Assess clinical benefit within two weeks. ^c^ Corticosteroids have no proven antiemetic effect in advance cancer, but must be used in case of brain metastases-related oedema or malignant bowel obstruction. Based on Lefroy J et al. “Guidelines: the do’s, don’ts and don’t knows of feedback for clinical education”.

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
