# Peer review of "Best Supportive Care of the Patient with Oesophageal Cancer"

_cancers, 2022, doi:10.3390/cancers14246268_

Round 1

Reviewer 1 Report

Dear Authors,

congratulation to your comprehensive review. I have only on minor suggestion/question, which could be changed:

"Xtampza (branded oxycodone ER) and Kadian (branded morphine SR)"

Are these drugs only available in Portugal? Anyhow, I would suggest to delete the brand name..

Good luck with your manuscript,

Gudrun Kreye

Author Response

Dear Dr. Gudrun Kreye,

Thank you for your kind comments and suggestion.

Guyer DL et al. (Ref. 41), who have USA affiliations, have mentioned Xtampza and Kadian as examples of long acting formulations that can obviate the need to swallow pills: “(…) Xtampza (branded oxycodone ER) and Kadian (branded morphine SR) are specific abuse-deterrent capsules that have the option of being opened and the microsphere contents can be swallowed with applesauce or administered through a feeding tube.”

As we have written in the manuscript “(these [Xtampza and Kadian] are not yet available in all countries, such as ours [Portugal])”. These drugs have FDA approval, but we did not find any equivalent alternatives in Europe. There were other prolonged-release morphine sulfate capsules (as Ethirphin), that could not be open, and raised safe concerns.

Considering this, we rephrased this part of the manuscript to:

“Regarding long-acting (sustained or extended-release) opioids, there are few alternatives to tablets, such as fentanyl and buprenorphine transdermal systems, and some specific oxycodone and morphine sulfate capsules that can be opened so the content can be swallowed or administered through a feeding tube [41]. Yet, these options are not available in every country or institution.”

Reviewer 2 Report

REVIEW Best Supportive Care of the Patient with Oesophageal Cancer

Abstract.

Why only search in pub med?

When you write records – what do you mean? Please be specific.

Introduction

Normally one would write …(22%) and not …(by 22,0%),

Consider including known symptoms of oesophageal cancer

Consider including if men or female prevalence, is there are trend between ethnicitiy e.g.,

Aim ok

Methods

Authors – please state how many of the authors who performed the search

Why only one database – is it a systematic review if you only use one database?

Who did you include the papers? Please elaborate.

What about langues?

Exclusion criteria?

This section needs more text to explain the process.

How many times did you search for articles?

Who decided which article to include?

Consider proving a table to show how you rated the included articles.

Results

Some of the text should be moved to the method section.

What do you mean with records?

Please don’t discuss your findings in the result section, Instead, provide a discussion section.

Author Response

We thank you for the careful review and comments, which we took into consideration to re-write the revised version of the manuscript.

In general, we were not able (nor intended) to perform a strictly systematic review, due to the scope of the subject “supportive care of oesophageal cancer”, and due to lack of high level evidence from clinical trials on this matter. Instead, as stated in the manuscript, authors performed an evidence-based review, including, not only clinical trials or meta-analysis, but also searching for relevant clinical practice guidelines, reviews and other original articles. Having this in mind, unfortunately, it may not be possible to reply all questions and suggestions.

Please see the attachment for more detailed ou direct responses to each comment.

Reviewer 3 Report

The authors have presented significant research displaying the best supportive care for patients with oesophageal cancer. The review is scientific and addressed one of the clinical issues related to oesophageal cancer management.  This study would be beneficial to future researchers in elucidating a better prognosis for cancer management. However, why do authors have created paragraphs in the conclusion section? Kindly rewrite the conclusion section in an appropriate manner. Altogether, this manuscript can be accepted in its present form with some minor corrections.

Author Response

Thank you for your considerations.

In reply to the reviewer’s suggestion to rewrite the conclusion section, this is the conclusion paragraph of the revised manuscript:

“Oesophageal cancer has a great personal and global impact. Often it presents as an advanced disease, with patients requiring best supportive care. However, apart from trials on the use of stents, there is a lack of high level evidence regarding the several treatment options available. More well-designed trials and observational studies are needed to determine which approaches achieve the greatest benefit for these patients. Optimal management is not yet standardized, and best supportive care decisions should be discussed on a case-by-case basis by multidisciplinary teams.”

Round 2

Reviewer 2 Report

The authors have modified the paper.